# Location of Multiple Types of Faults in Active Distribution Networks Considering Synchronization of Power Supply Area Data

**Gang Ren [1], Xianguang Zha [2], Bing Jiang [3],***, **Xiaoli Hu [1], Junjun Xu [3] and Kai Tao [3]**

[1] Skill Training Center of State Grid Jiangsu Electric Power Co., Ltd., Suzhou 215004, China
[2] State Grid Jiangsu Electric Power Co., Ltd., Nanjing 210024, China
[3] College of Automation & College of Artificial Intelligence, Nanjing University of Posts and Telecommunications, Nanjing 210023, China
*   Correspondence: jiangb@njupt.edu.cn

**Abstract:** When a short circuit occurs in the power supply area of a distribution network with a high-permeability distributed generation, the line current will increase, the voltage will drop sharply, and the fault characteristics will be more complex. Therefore, the automatic, quick, and accurate location of fault sections is of great significance to the reliability of power supply. In order to prevent large-scale power outages in the power supply area caused by the failure of feeders and transformers, this paper proposes a novel method to locate fault sections in active distribution networks by taking into account the data of the power supply area. On the basis of the synchronization of calculated and measured time and the observability of the fault state, a limited number of intelligent terminals are reasonably arranged in the distribution network feeder and power supply area. Additionally, the fault location equation is established based on the three-phase voltage change values of the nodes before and after the fault collected by intelligent terminals, so that the fault section is determined by comparing the residuals. Finally, the proposed method is verified by the improved IEEE 37-node and IEEE 123-node three-phase distribution networks. The results show that it has high accuracy in locating fault sections in multiple fault scenarios.

**Keywords:** active distribution networks; distributed generation; fault section location; intelligent terminal; data synchronization

## 1. Introduction

Distribution networks play a crucial role in national energy strategies. In order to promote the widespread grid connection of clean energy and ensure the distribution system's safe, dependable, and cost-effective functioning, it is essential to realize the panoramic perception of the region of the low-voltage distribution station [1].

The distribution system is responsible for providing power directly to consumers, but its complicated operational environment is prone to malfunctions. Among all power system, 80% of faults are related to the distribution network, and more than half of these faults are single-phase-to-ground faults [2]. A line failure will have a severe impact on the transformer as well as the power supply of its downstream nodes. When a ground fault occurs close to a transformer, it could result in a transformer discharge at the neutral point, which would further cause a series of complex faults such as transformer gap breakdown, transformer tripping, and gap current transformer damage [3]. This would eventually lead to a significant power outage in the power supply area. In order to increase the reliability of the power supply [4] and restore the power supply as soon as possible, it is essential to swiftly locate the fault section and fix the fault [5]. However, distribution network faults do not always have obvious characteristics. Most branch lines in the distribution network are short. Large amounts of distributed resources are being concealed in the grid at the same

time [6]. Electric vehicles are one example of a variable load [7]. The location of distribution network faults has proven to be very difficult to determine as a result of these uncertainties and the features of distribution networks.

The main methods for determining fault location are the steady-state method, transient method, the traveling wave method, and the artificial intelligence algorithm [8]. Among them, the steady-state method uses the amplitude and phase relationship of the power frequency voltage and current in the steady state to judge a fault after it occurs [9]. Reference [10] established the zero-sequence impedance model of the distribution network to locate the fault interval according to the zero-sequence fault current characteristics at the nodes. Reference [11] assumed that the fault resistance is purely resistive and does not consume reactive power, and established a quadratic equation with the fault distance as the unknown variable for the location of different fault types. The transient method extracts the zero-sequence current, power, and other features of the fundamental wave or harmonics by means of signal processing technology. Reference [12] proposes a ground fault discrimination method in low-current grounding systems by comparing the relationship between the magnitude and phase angle of the zero-sequence current at the fault outlet and the neutral point in the fault zero-sequence equivalent network. Reference [13] judges the fault location according to the appearance of the first peak value of current and voltage, but does not mention the application effect in a three-phase network. The distribution network's high density of branches and short lines makes it challenging for the traveling wave method to be effectively applicable [14]. Reference [15], based on the double-terminal traveling wave principle, proposed the fault branch judgment principle of three-terminal and multi-terminal transmission lines to accurately determine branch line faults, T-node faults, and line faults between T-nodes. Reference [16] analyzed the particularity of the fault traveling wave signal in the distribution network on the basis of Reference [14], who established a fault branch search matrix based on the arrival time of the multi-terminal initial traveling wave, and compares the changes of the matrix elements before and after the fault to locate the fault section. Reference [17] simulated the injection of DC pulse signals for positioning after the line is disconnected from the grid. These methods are not subject to factors such as operation mode, fault type, and current sensor saturation, but conditions such as high-speed sampling systems need to be taken into account. In most cases, the effect of fault type and distributed generation (DG) does not need to be considered. The artificial intelligence algorithm, such as that suggested in Reference [18], needs a lot of training data to finish learning the eigenvalues. Nevertheless, there are not enough training data or training samples, so the algorithm will end up in an optimal local situation [19]. Reference [20] introduces a method of using CNN to analyze the bus voltage measurement for fault location in a targeted manner, and it can ensure a certain fault section location ability when some bus voltage data are missing.

In the distribution network, intelligent terminals should be logically configured to minimize the negative effects of faults [21], as they can provide high-precision node voltage, current amplitude, and phase angle measurement information in real-time and synchronously for the convenience of distribution network fault location [22,23]. Reference [24] proposed a fault location algorithm utilizing intelligent terminals without using line parameters, but it does not apply to small current ground faults. Reference [25] modeled the fault node using the equivalent injection current method, but its accuracy is constrained by the equivalent model of distributed power. Reference [26] calculated the voltage index $V_{VI}^F$ and performed the measurement $V_{sen}^F$ at the relay point. By comparing the magnitudes of $V_{VI}^F$ and $V_{sen}^F$, it can know whether the fault is located before or after the intersection point.

In general, the massive branches and short lines of a distribution network, along with its own fault characteristics, are constraints on its ability to quickly locate the fault section. In light of the aforementioned issues, this paper analyzes the distribution network state observability and optimizes the configuration of intelligent terminals by performing compatibility processing on the time asynchrony of measurement data brought on by the diversity of equipment. On this foundation, a distribution system short-circuit fault

equivalent model was developed. The virtual fault state variables are solved based on the sparse measurement data from meters installed in the feeder and distribution power supply area. The linear least squares residuals are then compared to locate the fault section. Finally, the trend of the three-phase line voltage change in the fault section is used to determine the fault phase.

In order to verify the effect of the algorithm in the actual scene, we tested it on the improved IEEE-37 node system and IEEE-123 node system. According to the test results, we draw the following conclusions:

1. The algorithm used in this paper is relatively simple to use. Additionally, the algorithm has a higher fault location accuracy in high-permeability distribution networks;
2. The algorithm has a high utilization rate of measurement data, reduces the number of intelligent terminals, and has higher economic efficiency;
3. The algorithm can discriminate different types of fault categories with high accuracy.

The content of this paper is organized as follows: Section 2 introduces the use of multiple types of intelligent terminals for the observation analysis of distribution network fault states. Section 3 introduces the fault location algorithm for a three-phase unbalanced distribution network. Section 4 verifies the feasibility of the algorithm by using an improved example of IEEE-37 node distribution network. Section 5 concludes this paper.

## 2. Observation Analysis of Distribution Network Fault States Based on Multi-Type Intelligent Terminals

In order to fully monitor the faults of the distribution network, intelligent terminals must be configured at the corresponding bus of the distribution system. It is extremely important to obtain higher fault location accuracy through limited intelligent terminals.

### 2.1. Intelligent Terminal Measurement Time Synchronization

Due to the different delays in the links of transmission, communication, reception, storage, and application caused by the measurement data obtained by multi-type intelligent terminals, their direct use may result in information incompatibility, which will negatively impact the observability of faults. By examining the correlation coefficients of various types of measurement data and their sampling time using the signal correlation theory, the base sampling time of the measurement system is established, allowing the real-time measurement data of the intelligent terminal to be synchronized at a specific time section. The Pearson correlation coefficient, which measures the time synchronization relationship of different measurement signals, is:

$$\rho(t_1, t_2 - \delta) = \frac{C(t_1, t_2 - \delta)}{\sqrt{C(t_1, t_1) \cdot C(t_2 - \delta, t_2 - \delta)}} \tag{1}$$

where $C$ is the covariance; $t_1$ is the measurement data without the time scale, and the representative device of this type of intelligent terminal is FTU; $t_2$ is the measurement data with time scale, $\delta$ is the measurement time series, and the representative device of this type is μPMU.

Assume that the FTU measurement upload rate is $f_S$, and the μPMU data upload rate is $f_\mu$. In a given time $T$, there are $n_s = T \cdot f_s$ sets of FTU measurement data:

$$z_F = [z_{F,1}, z_{F,2}, \cdots, z_{F,n_s}] \tag{2}$$

In the same way, there are $n_\mu = T \cdot f_\mu$ sets of μPMU measurement data:

$$z_\mu = [z_{\mu,1}, z_{\mu,2}, \cdots, z_{\mu,n_\mu}] \tag{3}$$

Then, calculate the Pearson correlation coefficient of each element in $z_F$ and $z_\mu$, arrange the correlation coefficient $\rho$ according to the time series, take the μPMU sampling time

corresponding to the maximum value as the reference time of the multi-intelligent terminal measurement system, and calculate the FTU measurement and this time delay error $\varepsilon_t$.

$$\varepsilon_{\mathrm{t}} = k t_{\mathrm{d,F}} \tag{4}$$

where $k$ is the rate of change of the measurand, $t_{\mathrm{d,F}}$ is the deviation between the base time and the state estimation calculation time, which can be considered to obey the following probability density function.

$$f_t(t_{\mathrm{d,F}}) = \frac{1}{\sigma_t \sqrt{2\pi}} \mathrm{e}^{-(t_{\mathrm{d}}-t)^2/2\sigma_t^2} \tag{5}$$

The symbol $\sigma_t$ is the standard deviation of the probability density function.

### 2.2. Failure State Observability

When considering the measurement noise, the relationship between the measurement variable and the state variable in the distribution network is:

$$z = H(x) + v \tag{6}$$

where $z$, $v$, $x$, and $H$ are the measurement vector, measurement noise vector, state vector, and measurement function vector, respectively.

From a mathematical point of view, in order to ensure the observability of the system and the accurate determination of the subsequent fault section location, the configuration of the intelligent terminal should make the $H$ matrix full rank, that is, the number of measurement linear correlations is the smallest. At the same time, the measurement vector dimension must be larger than the state vector dimension:

$$\begin{aligned} \mathrm{R}(M^T M) &= \dim(x) \\ \dim(z) &> \dim(x) \end{aligned} \tag{7}$$

where $\mathrm{R}(\cdot)$ represents the order of the square matrix, and $\dim(\cdot)$ represents the dimension of the vector.

It should be noted that a ground fault in the vicinity of a distribution transformer may lead to the transformer neutral point discharge, which would further cause a series of complex faults such as transformer gap breakdown, transformer tripping, and gap current transformer damage. The power supply of the corresponding power supply area can be directly determined by monitoring the transformer's operating state. As a result, in this paper, intelligent terminals are installed on the low-voltage side of the transformer. The distribution network is primarily radial at the same time. Considering that the configuration nodes of intelligent terminals, especially μPMU measurement equipment and its adjacent nodes, are all observable nodes [27,28], the nodes with more outgoing lines should be selected as configuration nodes as much as possible.

## 3. Three-Phase Unbalanced Distribution Network Fault Location Algorithm

### 3.1. Distribution Network Model in Normal and Faulty Operating States

Figure 1 depicts the three-phase line's type equivalent admittance model in normal operation between any two nodes $i\,j$ in the three-phase distribution network. $Y_{ij}{}^{aa}$ represents the line admittance of the a-phase between node $i$ and $j$; $Y_{ii}{}^{ab}$ represents the mutual admittance between the a-phase and b-phase; $Y_i{}^{aa}$ represents the a-phase-to-ground admittance of node $i$; $Y_i{}^{ab}$ represents the admittance between the a-phase and b-phase at node $i$. Therefore, taking nodes $i$ and $j$ as an example, the network should conform to the following equation when it operates normally:

$$I^\varphi = Y^\varphi U^\varphi \tag{8}$$

where $I^\varphi$ is the three-phase injection current of the node, $Y^\varphi$ is the node admittance matrix, and $U^\varphi$ is the three-phase voltage of the node.

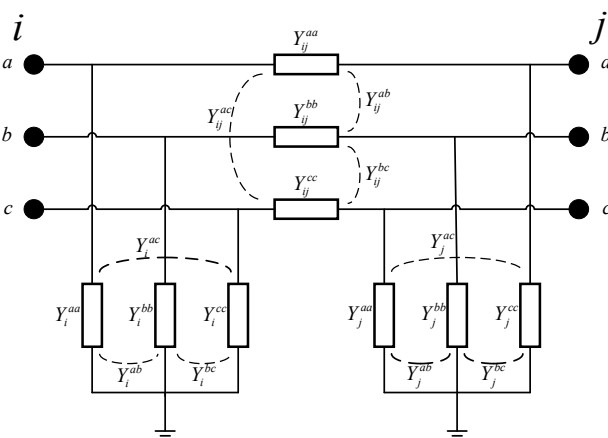

**Figure 1.** Normal model of distribution networks.

The failure of a distribution line can be compared to expanding the original model by a virtual node and injecting current into the faulty line from this virtual node. As shown in Figure 2, when the c-phase line between nodes *i* and *j* is faulty, the equivalent fault model can be changed to have one equivalent grounding current source at the c-phase at both nodes in order to lessen the effect of virtual nodes on subsequent calculations. When the phase line fails, the equivalent current source will be non-zero, whereas when the system operates normally, the injection current of the equivalent current source to the node is zero.

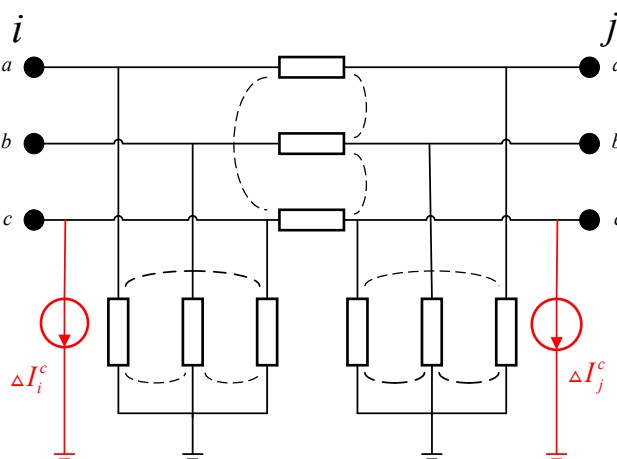

**Figure 2.** Single-phase-to-ground fault model of distribution networks.

When the line fails, the three-phase injection current of the node and the three-phase voltage of the node have the following relationship:

$$I^{\varphi\prime} = Y^\varphi U^{\varphi\prime} \tag{9}$$

where $I^{\varphi\prime}$ and $U^{\varphi\prime}$ are the node current and voltage after the fault occurs.

The different wiring modes of the transformer also affect the short-circuit fault characteristics differently because it is the main device for connecting various voltage levels. The common wiring mode of a low-voltage distribution transformer is Dyn11, that is, the voltage phase angle changes by 30° when the primary side phase voltage is converted into the secondary side line voltage. When a short-circuit fault occurs in the medium-voltage side line of the distribution network, the voltage sag value caused by it propagates to the low-voltage side and is not directly propagated according to the transformation ratio. The

change in fault characteristics is shown in Table 1. Conversely, when a short-circuit fault occurs on the low-voltage side of the distribution network, due to the large leakage reactance of the transformer, the impact on the primary side can be ignored after the transformer is isolated. The propagation effect of the transformers with other wiring modes on the system feeder fault is shown in [29].

**Table 1.** Propagation effect of the transformer on fault characteristics.

| MV Side Fault Type | MV Side Voltage Characteristics | Whether LV Side Has Same Fault Characteristics as MV Side |
|---|---|---|
| Single-phase-to-ground | The voltage of fault phase is lowered and that of the non-fault phase is raised | No |
| Two-phase-short-circuit | The voltage of fault phase is lowered and that of the non-fault phase is raised | No |
| Three-phase-short-circuit | The voltages of all the phases are raised | Yes |

### 3.2. Fault Location Algorithm

(1)  *Fault location start criterion*

In this paper, the mutation detection algorithm [30] will be used to determine the moment when the fault occurs. For the voltage $U_M$ collected by the intelligent terminal on the low-voltage side of the distribution transformer and at the key nodes of the system, when the voltage sampling value change $\Delta U_M$ in the adjacent cycle is set to the limit three times in a row, it is considered that the previous sampling that exceeded the limit for the first time is the time of failure. The out-of-limit expression is:

$$|\Delta U_M(k+N) - \Delta U_M(k)| \geq k_f U_{dz} \tag{10}$$

where $k_f$ is the constant value adjustment coefficient of mutation (usually $k_f = 1$); $U_{dz}$ is the starting value of mutation.

(2)  *Fault location and fault phase identification criteria*

In case the line parameters and topology of the distribution network are known, the fault state is used as the variable to be estimated to construct the measurement equation:

$$z' = H(x) + v \tag{11}$$

where $z'$ is the fault measurement vector and $x$ is the fault state vector.

When a short-circuit fault occurs in the distribution network, it can be seen from the three-phase unbalanced distribution network operation model that the changes in node voltage and current due to the fault satisfy the following equation:

$$\Delta I^\varphi = Y^\varphi \Delta U^\varphi \tag{12}$$

where $\Delta I^\varphi$ is the node equivalent three-phase fault current and $\Delta U^\varphi$ is the node voltage variation. Reversing Equation (11), take the fault equivalent current as the state quantity, and take the node voltage measurement as the quantity measurement:

$$\begin{bmatrix} \Delta U_1^\varphi \\ \Delta U_2^\varphi \\ \vdots \\ \Delta U_m^\varphi \end{bmatrix} = H \cdot \Delta I^\varphi \tag{13}$$

where $m$ is the number of intelligent terminal devices in the distribution network.

Due to the existence of DG in the distribution network, considering that it is difficult to estimate its effect on the fault current during the fault period, the voltage change value of

the node where it is located is taken as the fault state quantity, and Equation (13) is further transformed into:

$$
\begin{bmatrix} \Delta U_1^\varphi \\ \Delta U_2^\varphi \\ \vdots \\ \Delta U_m^\varphi \end{bmatrix} = H' \cdot \begin{bmatrix} \Delta U_{DG1}^\varphi \\ \Delta U_{DG2}^\varphi \\ \vdots \\ \Delta U_{DG\,s}^\varphi \\ \Delta I^{\varphi'} \end{bmatrix} \tag{14}
$$

where $s$ is the number of DGs in the distribution network and $\Delta I^{\varphi'}$ is the fault injection current of $n$-$s$ nodes.

A common method for solving fault state variables is the least squares estimation algorithm, which establishes the objective function according to the weighted least squares principle:

$$
J(x) = (z - Hx)^T W(z - Hx) \tag{15}
$$

Take the derivative of the objective function and take it to zero, that is:

$$
\frac{\partial J(x)}{\partial x} = 0 \tag{16}
$$

Solve the above equation to get the x value and write it in matrix form:

$$
\hat{x} = \left( H^T R^{-1} H \right)^{-1} H^T R^{-1} z \tag{17}
$$

where $\hat{x}$ is the estimated value of the state quantity; $H^T R^{-1} H$ is the information matrix; $W = R^{-1}$ is the measurement error variance matrix, and its elements are:

$$
R_i^{-1} = \frac{1}{\sigma_i^2} \tag{18}
$$

where $\sigma$ is the standard deviation of the measurement error.

When the node $k$ is assumed to be a virtual fault node, variable $\Delta I_k^\varphi$ in $\Delta I^{\varphi'}$ in Equation (14) is taken as a separate quantity to be calculated, and the coefficient matrix **H** is adjusted to obtain the virtual fault state equation:

$$
\begin{bmatrix} \Delta U_1^\varphi \\ \Delta U_2^\varphi \\ \vdots \\ \Delta U_m^\varphi \end{bmatrix} = H'' \cdot \begin{bmatrix} \Delta U_{DG1}^\varphi \\ \Delta U_{DG2}^\varphi \\ \vdots \\ \Delta U_{DG\,s}^\varphi \\ \Delta I_k^\varphi \end{bmatrix} \tag{19}
$$

Since $m > s + 1$, Equation (19) is an overdetermined equation. The least squares method is used to calculate the residual value of the equation, which is small when the virtual fault point agrees with the actual fault point, and large when the virtual fault point differs from the actual fault point. The fault section can then be identified.

The voltage of the fault phase where the ground fault occurs will decrease, while the voltage of the non-fault phase will increase, allowing for the location of the fault section. The variations in each phase's voltage in the fault section can be used to identify the fault phase. This indicates that the fault phase is the phase whose voltage decreases while the voltage of the other two phases increases. The fault phase is also the two phases or the three phases if the voltage in those phases decreases. At the same time, a suitable threshold should be set when determining whether the voltage increases or decreases due to measurement errors. Therefore, the location of the distribution network fault sections considering the data of the distribution power supply area proposed in this paper is divided into the five following steps.

(1) The coefficient matrix of the state equation is determined according to the distribution network structure, line parameters, and the configuration of the intelligent terminal.

(2) Using the fault detection algorithm to determine the moment of fault occurrence;

(3) The fault injection current is then calculated using the voltage measurement values before and after the fault occurs;

(4) Use the least squares method to calculate the residual error of the virtual fault state equation. Then, identify faulty segments based on residual size;

(5) Determine the fault type according to the three-phase voltage change of the fault section.

The basic process is shown in Figure 3.

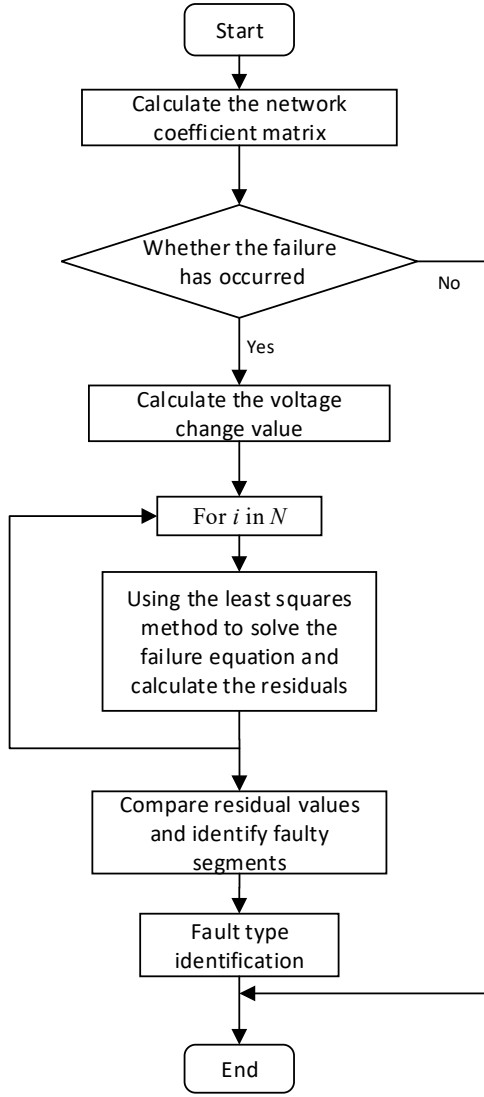

**Figure 3.** The flowchart of the fault section location and fault phase identification.

## 4. Case Study

### 4.1. The Improved IEEE-37 Node Distribution Network

(1) *Introduction to the distribution network*

Based on the improved IEEE-37 node three-phase distribution network, the simulation test was carried out, and its topology is shown in Figure 4. The system voltage level is 4.8 kV, the total powers of the three-phase load are 727 + j357 kV·A, 639 + j314 kV·A and 1091 + j530 kV·A, respectively. For the active and reactive power of each node of the distribution network, the connection method, the three-phase parameter, and reactive

power compensation, see [31]. In order to verify the applicability of the proposed fault section location method in the active distribution network within the distribution power supply area, distributed power sources with a rated power of 0.5 MW and a rated power of 0.1 MW are connected to nodes 21 and 28, respectively. At the same time, two transformers are installed at nodes 13 and 36, with a transformation ratio of 4.8 kV/0.4 kV, and the Dyn11 wiring mode is used. The load power of the power supply area is shown in Table 2.

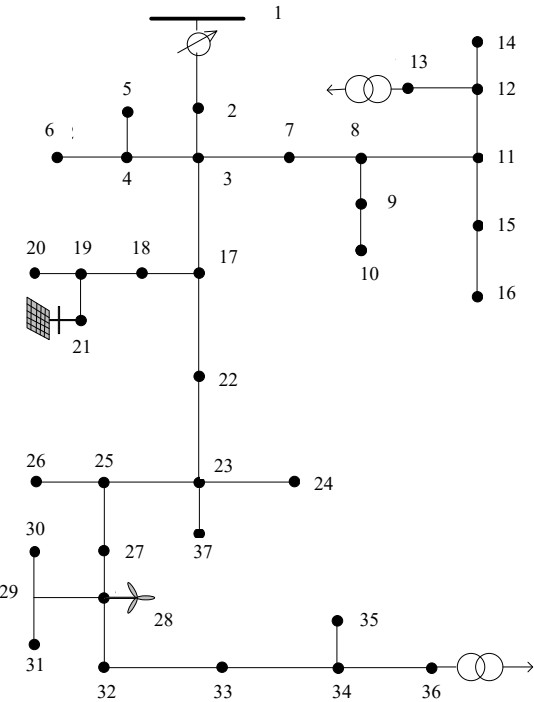

**Figure 4.** The improved IEEE-37 distribution network.

**Table 2.** The load power of the power supply area.

| Node No. | Complex Power/kV·A | | |
| --- | --- | --- | --- |
| | **A-Phase** | **B-Phase** | **C-Phase** |
| 13 | 0 | 140 + 70 j | 21 + 10 j |
| 36 | 0 | 0 | 42 + 21 j |

According to the network fault observability analysis, considering the location accuracy requirements of the fault section of the distribution network, intelligent terminals FTU and μPMU are configured at key nodes 3, 6, 10, 16, 19, 27, 29, 30, and 32 and nodes 13 and 36 that are connected to the transformer. Among them, the measurement error of the node voltage magnitude measured in real-time by FTU obeys a normal distribution with a mean value of zero and a standard deviation of 1% of the true value; the voltage magnitude error measured by μPMU in real-time obeys a normal distribution with a mean value of zero and a standard deviation of 1% of the true value. Additionally, the voltage angle measurement error follows a normal distribution with a mean of zero and a standard deviation of ±0.001°.

(2)  *Analysis of time synchronization of intelligent terminal measurement*

Table 3 shows the calculation results of the correlation degree of measurement data of multiple types of intelligent terminals. Taking some FTU and μPMU measurement data as an example, the maximum value of each row of data almost appears in the fifth column. Based on the correlation degree theory in Section 2.1, it is easy to see that the measurement delay error of the FTU measurement data is set to obey a normal distribution

with a standard deviation of 0.1/s and a mean value of 0, thus ensuring the synchronization and validity of the measurement data of multiple types of intelligent terminals.

**Table 3.** The results of multi-type intelligent terminal measurement data correlation.

| FTU Sampling Time | Correlation Degree of Measurement Data of μPMU ($\rho_{FP}$) | | | | | |
|---|---|---|---|---|---|---|
| | $t = 0.02$ s | $t = 0.04$ s | $t = 0.06$ s | $t = 0.08$ s | $t = 0.1$ s | $t = 0.12$ s |
| 24 | 0.84 | 0.48 | 0.62 | 0.74 | 0.98 | 0.54 |
| 25.2 | 0.38 | 0.15 | 0.67 | 0.57 | 0.72 | 0.60 |
| 26.4 | 0.85 | 0.25 | 0.81 | 0.64 | 0.89 | 0.72 |
| 27.6 | 0.58 | 0.19 | 0.41 | 0.74 | 0.81 | 0.80 |
| 28.8 | 0.90 | 0.51 | 0.35 | 0.57 | 0.93 | 0.35 |

(3) *Analysis of fault propagation characteristics in distribution power supply area*

This paper places the b-phase ground fault at the start of lines 12–13 and downstream from the node 13 transformer in order to analyze the fault propagation characteristics of the distribution power supply area. The simulation lasted 0.1 s in total. The fault duration was 0.03 s if the fault occurred in 0.04 s. Figures 5 and 6 depict the downstream power supply area (low voltage side) of node 13 and the fault voltage characteristics of node 12 (medium voltage side), respectively. It can be seen from Figures 5 and 6 that when a single-phase-to-ground fault occurs on the medium voltage side, the fault phase voltage on the medium voltage side decreases, and the non-fault phase voltage increases. After it propagates to the low voltage side of the Dyn11 transformer, both the fault phase and lagging phase voltages reduce, which satisfies the propagation relationship of the distribution transformer to fault characteristics described in Section 3.1. It can be seen from Figures 7 and 8 that when a single-phase-to-ground fault occurs on the low-voltage side of the power supply area, because the transformer has a certain electrical isolation effect, it has little influence on the voltage of the remaining nodes of the distribution network system, as well as the low-voltage side of the actual project. The low-voltage side should be tripped directly after a single-phase-to-ground fault to minimize the impact on the overall voltage level of the system.

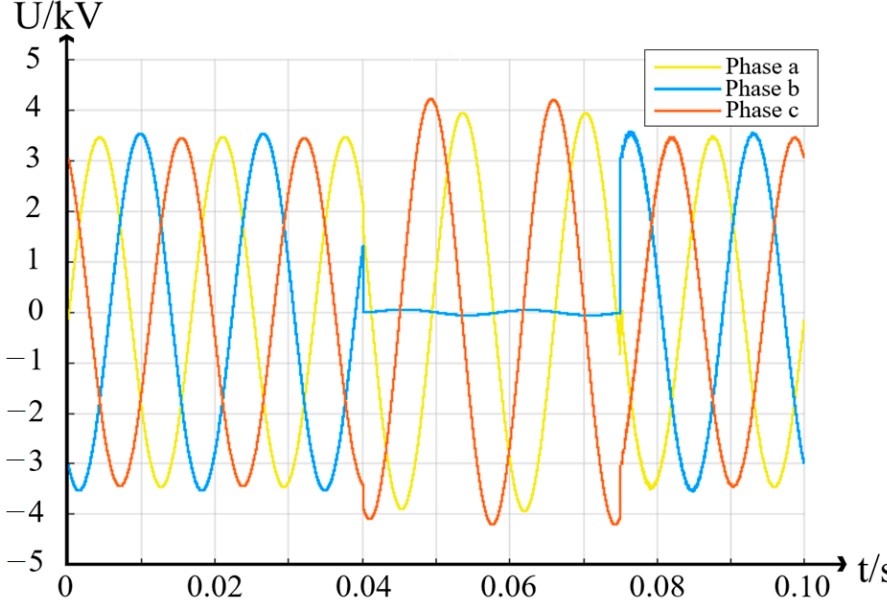

**Figure 5.** Voltage characteristics of the MV side when the MV side fails.

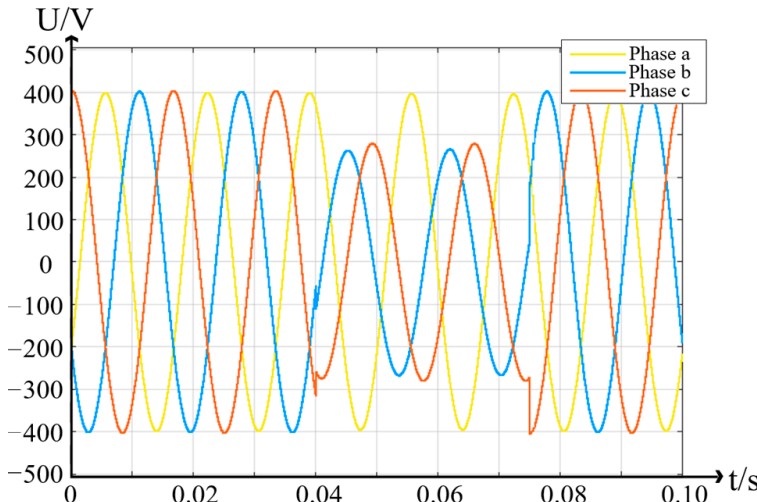

**Figure 6.** Voltage characteristics of the LV side when the MV side fails.

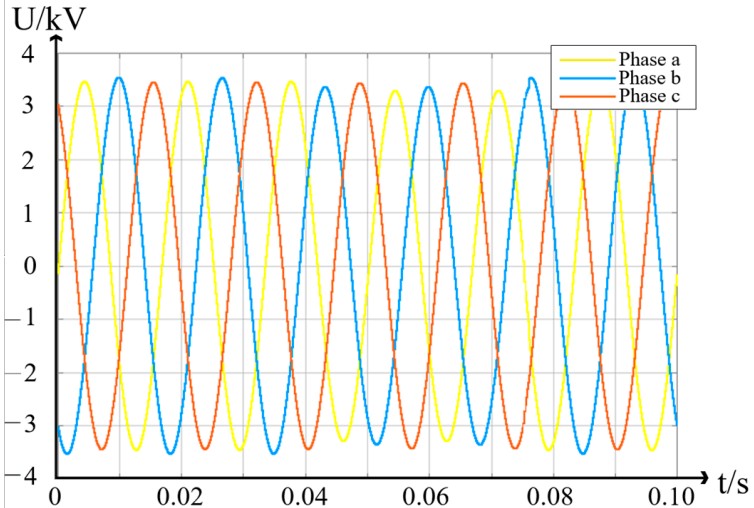

**Figure 7.** Voltage characteristics of the MV side when the LV side fails.

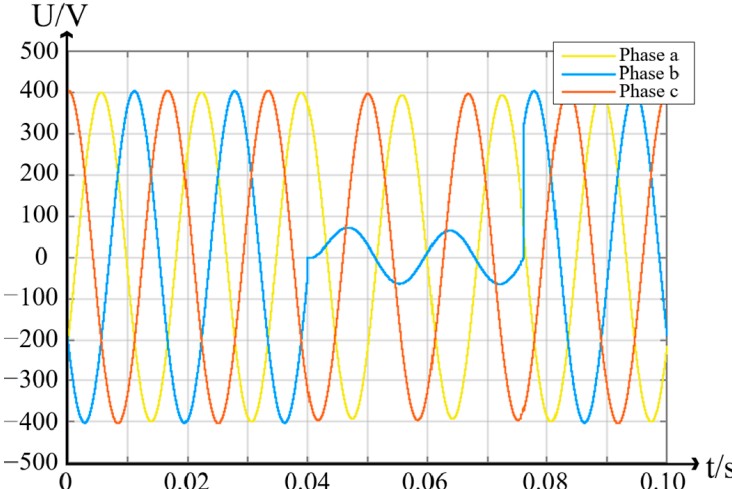

**Figure 8.** Voltage characteristics of the LV side when the LV side fails.

The residuals of the virtual fault state equation at various nodes are shown in Figure 6 in order to demonstrate the viability of the proposed fault section location method. It is

assumed that a short circuit or ground fault occurs at branches 8–11 of the improved IEEE-37 distribution network. In Figures 9 and 10, the horizontal coordinate represents the node number, and the vertical coordinate represents the residual of the equation. The yellow, green, and red colors represent the three phases, respectively. The grounding impedance corresponding to fault scenario 1 is 0.001 p.u., and the grounding impedance corresponding to fault scenario 2 is 1 p.u. In both scenarios, the smallest state estimation error is at node 8 or 11, which means that the fault occurs near bus 8 or 11, which is consistent with the actual fault location. Additionally, regardless of the value of the grounding impedance, the nodes in the area can be roughly divided into two categories according to the distribution range of the estimation error: one is located in the lower left part of the picture, representing the nodes with smaller state estimation errors, and these nodes are located on the faulty path 1–3–7–8–11. The estimation error corresponding to other types of nodes is larger because they are weakly connected to the fault path and less affected by the fault. At the same time, from the estimation error range in different scenarios, it can be seen that the smaller the grounding impedance is, the larger the residual error difference between the two types of nodes is, which ensures the high accuracy of the fault section location in the distribution network.

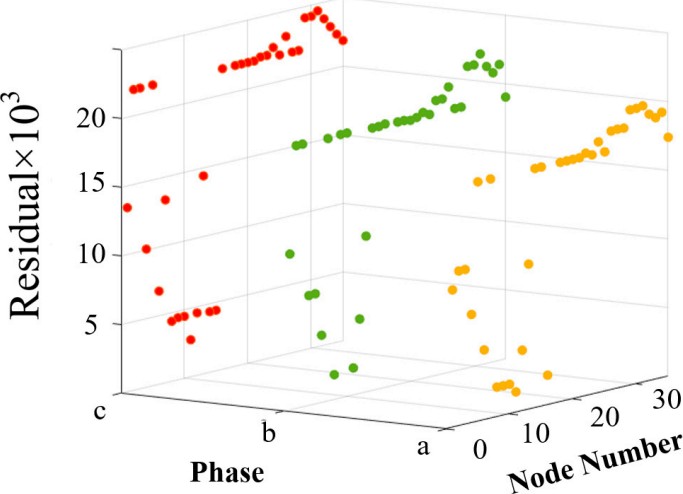

**Figure 9.** Residuals of virtual fault state equations at different nodes in scenario 1.

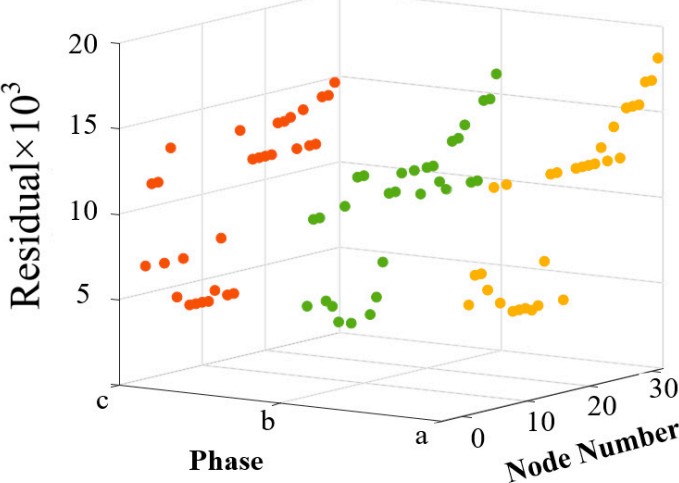

**Figure 10.** Residuals of virtual fault state equations at different nodes in scenario 2.

In order to further verify the accuracy of the proposed fault section location algorithm, four typical line terminals and two different distribution power supply areas were ran-

domly selected to set up short-circuit faults with different types and different grounding impedances in the improved IEEE-37 distribution network. The results are shown in Table 4. The fault section location accuracy index $P_t$ is defined as:

$$P_t = \frac{\lambda_t}{\lambda} \times 100\% \tag{20}$$

where $\lambda_t$ is the number of times for which the faulty section can be accurately identified upon $\lambda$ tests. It should be noted that the purpose of this paper is to locate the fault section of the distribution network. Considering that the length of the lines used in the example is short (most of them are approximately 100 m), the location point at the beginning or end of the branch falls within the correct results.

**Table 4.** Fault section location results in different lines and power supply areas in the IEEE-37 node network.

| Fault Line/Node | | $P_t$ | | | | | | | | |
|---|---|---|---|---|---|---|---|---|---|---|
| | | Single-Phase-to-Ground Short Circuit | | | Interphase Short Circuit | | | Three-Phase Short Circuit | | |
| | | 0 Ω | 100 Ω | 500 Ω | 0 Ω | 100 Ω | 1000 Ω | 0 Ω | 100 Ω | 1000 Ω |
| Fault Line | 11–15 | 100% | 100% | 100% | 100% | 100% | 100% | 100% | 100% | 100% |
| | 17–22 | 100% | 100% | (725) | 100% | 100% | 100% | 100% | 100% | 100% |
| | 32–33 | 100% | 100% | 98% | 100% | 100% | 100% | 100% | 100% | 100% |
| | 34–35 | 100% | 100% | 100% | 100% | 100% | 100% | 100% | 100% | 100% |
| Fault Node | Node 13 | 100% | 100% | 100% | 100% | 100% | 100% | 100% | 100% | 100% |
| | Node 36 | 100% | 100% | 92% | 100% | 100% | 100% | 100% | 100% | 100% |

Moreover, In order to study the influence of the DG permeability on the algorithm, the adaptability of the proposed single-phase-to-ground fault line selection method is verified under different DG permeability scenarios. The transition resistance is 300 Ω and 500 Ω, respectively, and the accuracy is calculated by 100 fault simulations. Table 5 shows the accuracy of the single-phase-to-ground fault line selection method when the DG permeability is 25% and 100%, respectively. In general, the proposed single-phase-to-ground fault line selection method for an active distribution network has high adaptability to the level of DG permeability. Because this method does not need to perform complex equivalent processing on DG, but adopts the idea of least squares estimation to solve the system state as a whole for fault line selection, which avoids the error caused by the inaccurate model in principle, and improves the fault line selection, as well as applicability and effectiveness of the method in high-permeability DG access scenarios.

**Table 5.** Comparison of the results under different DG permeability rates.

| Fault Node | $\eta$ | | | |
|---|---|---|---|---|
| | DG Permeability 25% | | DG Permeability 100% | |
| | 300 Ω | 500 Ω | 300 Ω | 500 Ω |
| Node (4, 5) | 100% | 100% | 100% | 100% |
| Node (7,8) | 100% | 100% | 100% | 100% |
| Node (11, 15) | 100% | 100% | 100% | 100% |
| Node (12, 13) | 100% | 100% | 100% | 100% |
| Node (17, 22) | 73% | 77% | 70% | 68% |
| Node (23, 25) | 100% | 100% | 100% | 100% |
| Node (29, 30) | 73% | 75% | 77% | 72% |
| Node (34, 35) | 100% | 100% | 100% | 100% |

### 4.2. The Improved IEEE-123 Node Distribution Network

In order to verify the application effect of the algorithm in the large-scale distribution network, this section will verify its performance in the IEEE-123 node system.

(1)    *Introduction to the large-scale distribution network*

The improved IEEE-123 node three-phase distribution network topology is shown in Figure 11. In this paper, four µPMUs are selected at nodes 1, 35, 76, and 97 to measure the node voltage phasors; the measurement is used to measure the branch flow power and node voltage amplitude. In addition, in order to obtain the more accurate power exchange of the upstream and downstream sub-regions, some FTU devices are installed to measure the branch flow power. Other nodes in the distribution network are pseudo-measurements, usually derived from short-term power predictions to provide the approximate values of active and reactive power injected into the nodes. The three-phase load type, active and reactive power, connection method of each node of the distribution network, and the three-phase parameters of the line and reactive power compensation are detailed in Reference [2]. In this paper, DGs are installed in the different phases of seven nodes in the network as shown in Table 6. Among them, the uncertain error of DG output obeys a normal distribution with a mean of zero and a standard deviation of 20% of the true value.

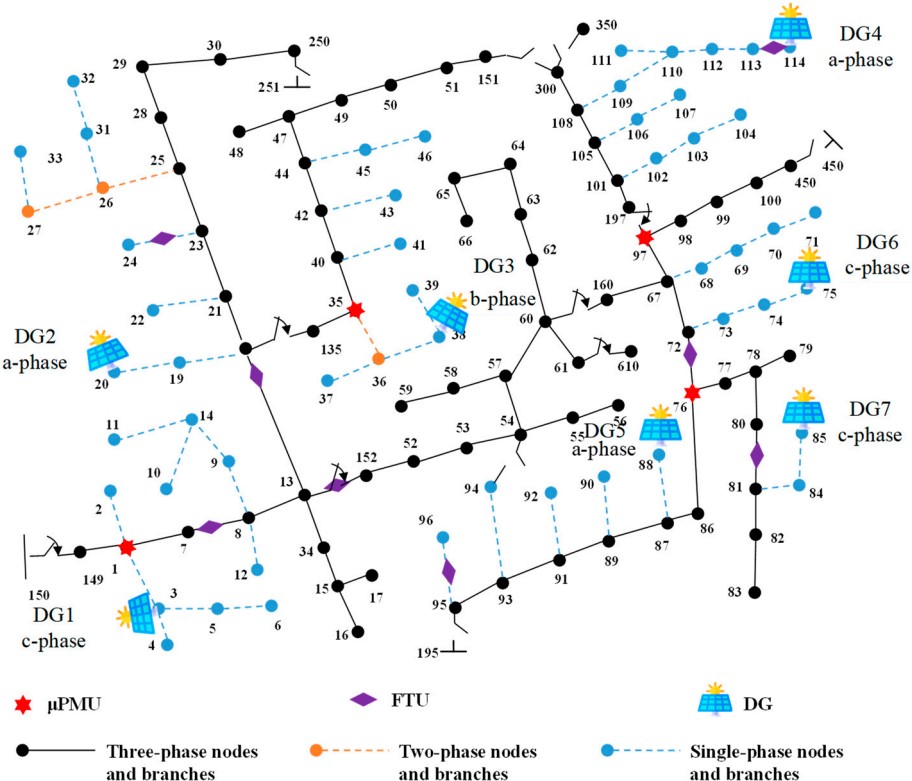

**Figure 11.** The improved IEEE-123 distribution network.

**Table 6.** Placement and capacity of DGs.

| DG No. | Deploy Node | Deploy Phase | P/kW |
| --- | --- | --- | --- |
| DG1 | 3 | C | 10 |
| DG2 | 20 | A | 15 |
| DG3 | 38 | B | 5 |
| DG4 | 114 | A | 10 |
| DG5 | 88 | A | 10 |
| DG6 | 75 | C | 10 |
| DG7 | 85 | C | 10 |

(2)   *Comparisons and analysis*

Since it has the same measurement time synchronization and similar fault propagation characteristics of intelligent terminals as the IEEE-37 node power saving system, the results are directly analyzed for the IEEE 123-node. The accuracy of the method proposed in this paper and the method proposed in Reference [2] are compared, and the results are shown in Table 7. The symbol 'A' in the table represents the algorithm proposed in this paper, while the symbol 'B' represents the algorithm proposed in Reference [2]. Additionally, Figure 12 shows the three-phase voltage magnitude and phase angle of the nodes near the fault location when the b-phase-to-ground short circuit occurs at nodes 8–9.

**Table 7.** Fault section location results in different lines and power supply areas in IEEE-123 node network.

| Fault Line | | | $P_t$ | | | | | | | | |
|---|---|---|---|---|---|---|---|---|---|---|---|
| | | | Single-Phase-to-Ground Short Circuit | | | Interphase Short Circuit | | | Three-Phase Short Circuit | | |
| | | | 0 Ω | 100 Ω | 500 Ω | 0 Ω | 100 Ω | 1 kΩ | 0 Ω | 100 Ω | 1 kΩ |
| Fault Line | 8–9 | A | 100% | 100% | 100% | 100% | 100% | 100% | 100% | 100% | 100% |
| | | B | 100% | 100% | 99% | 100% | 100% | 100% | 100% | 100% | 100% |
| | 67–72 | A | 100% | 100% | 98% | 100% | 100% | 99% | 100% | 100% | 100% |
| | | B | 100% | 100% | 99% | 100% | 100% | 99% | 100% | 100% | 99% |
| | 49–50 | A | 100% | 100% | 98% | 100% | 100% | 100% | 100% | 100% | 100% |
| | | B | 100% | 98% | 96% | 100% | 98% | 98% | 100% | 100% | 94% |
| | 64–65 | A | 100% | 100% | 99% | 100% | 100% | 100% | 100% | 100% | 100% |
| | | B | 100% | 94% | 92% | 100% | 96% | 94% | 100% | 98% | 94% |
| | 105–108 | A | 100% | 100% | 91% | 95% | 90% | 85% | 98% | 95% | 93% |
| | | B | 100% | 92% | 90% | 100% | 88% | 86% | 100% | 85% | 81% |

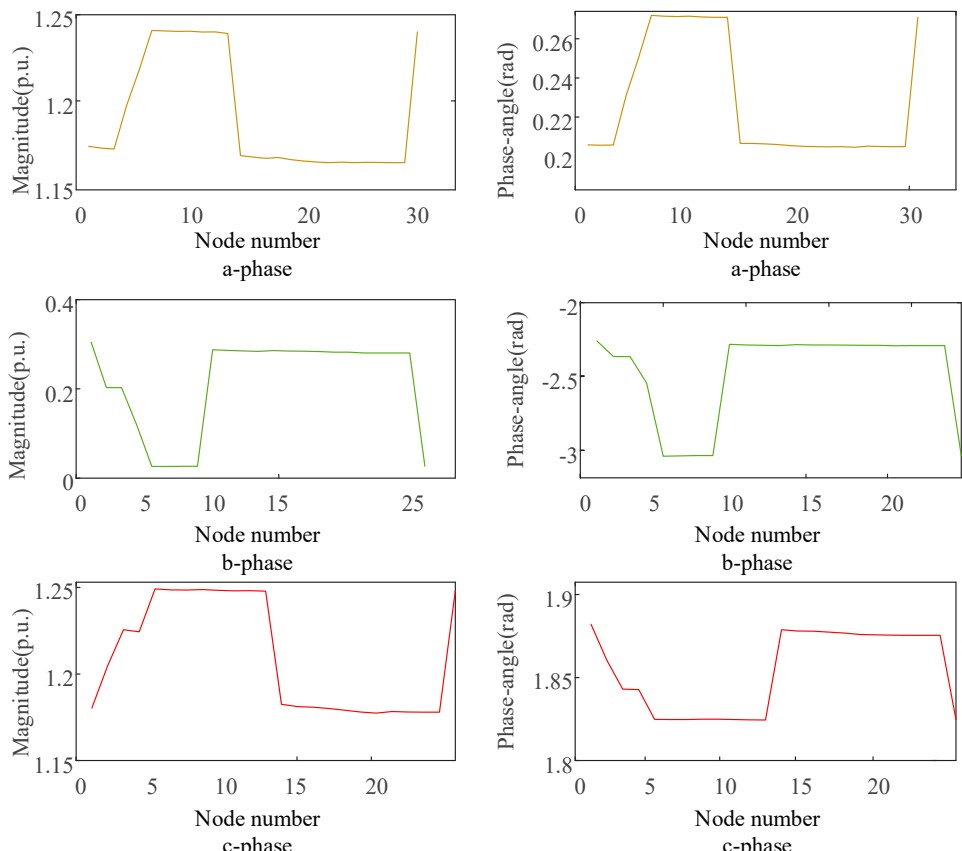

**Figure 12.** Partial three-phase voltage magnitude and phase angle of nodes near the fault location.

*4.3. Results*

Through the application of the fault location algorithm proposed in this paper in the IEEE-37 node and IEEE-123 node, the following results can be obtained.

(1)    *Results of the proposed method applied to IEEE-37 nodes*

It can be seen from Table 4 that when the tested line has an interphase short-circuit or three-phase short-circuit faults, the fault section location accuracy can reach 100%. When a single-phase-to-ground short circuit occurs in the test line and the transition resistance is less than 500 Ω, the accuracy of the fault section location is 100%. When the transition resistance of the single-phase-to-ground fault continues to increase to 500 Ω, there is the misjudgment of one line. The reason is that with the increase in the transition resistance value, the characteristic quantity of the system fault becomes smaller and smaller, and the fault section location accuracy reduces. In addition, when a single-phase-to-ground short circuit, phase-to-phase short circuit, or three-phase short circuit occur in the four distribution power supply areas in the example, the proposed fault location method can accurately locate the fault.

In the tested four typical lines and two distribution power supply areas, there are not only network trunk branches such as lines 17–22, but also network end branches such as 34–35, which indicate that the proposed fault section location method has high adaptability at different positions of the distribution network. The accuracy of the fault judgment results for branches 34–35 and the power supply area near node 36 shows that the method can effectively distinguish short branches with common nodes.

From Table 5, the proposed single-phase-to-ground fault line selection method for an active distribution network has high adaptability to the level of DG permeability. Because this method does not need to perform complex equivalent processing on DG, but adopts the idea of least squares estimation to solve the system state as a whole for fault line selection, it avoids the error caused by the inaccurate model in principle and improves the fault line selection as well as applicability and effectiveness of the method in high-permeability DG access scenarios.

(2)    *Results of the proposed method applied to IEEE-123 nodes*

It can be seen from Table 7 that both the fault location algorithms are not affected by the magnitude of the grounding impedance, and the location accuracy is high in most cases. The lines involved in the fault scenario set up in this paper include not only key trunk branches such as branches (67, 72), but also network end branches such as branches (49, 50); it includes both three-wire lines and single-phase lines such as branches (8, 9); including both overhead lines and cable lines (64, 65) which reflect the proposed high adaptability of the fault section location method in the large-scale three-phase distribution network. By comparing the method proposed in this paper with that proposed in Reference [2], it can be seen that the former has higher accuracy. This also means that the algorithm proposed in this paper is more effective in judging the fault category. It can be seen from Figure 12 that the changing trend is consistent with the fault propagation characteristics.

However, in Table 7, when a single-phase ground fault occurs at line (105, 108) and the grounding impedance is 0.001 p.u., the fault location accuracy is low. Although most of the fault section location results at this time indicate that the single-phase-to-ground fault occurs at line (105, 108), there are still some results that determine that the fault occurs at its adjacent line (108, 300). One reason is that the average length of the line in the example is short; another reason is that since node 300 has no load, the actual value of the node voltage is almost the same as that of node 108, resulting in a lack of distinction between its fault characteristics and adjacent nodes. Although the judgment result of the fault section cannot correctly indicate the fault line at this time, it is also close to the actual fault line which still has direct guiding significance for the automatic, quick, and accurate location of fault sections.

## 5. Conclusions

Aiming to grapple with the low fault location accuracy caused by the complex short-circuit fault characteristics in active distribution networks, this paper comprehensively considers the measurement data of intelligent terminals in the feeder and distribution power supply area, and proposes an algorithm for a fault section location in active distribution networks based on the solution of the virtual fault state value. The following conclusions are drawn:

(1) This paper analyzes the observability of faults in the distribution network. Intelligent terminals are installed at key nodes and low-voltage side nodes of transformers to measure the voltage drop value before and after the fault occurs. Additionally, the algorithm performs time synchronization processing on the measured data to provide data support for fault section location.

(2) The linear least squares method is used to quickly solve the virtual fault state equation, which has high accuracy for various short-circuit fault types and fault impedances.

(3) The proposed algorithm directly solves the voltage value of DG as the fault state quantity, which can remove the uncertainty of the power flow direction caused by the grid connection of DG, and has high applicability in the active distribution network with DG.

(4) According to the application of the proposed method in IEEE-37 nodes and IEEE-8500 nodes, this method can complete the fast and accurate location of faults. At the same time, the algorithm is also very accurate in identifying fault types. In the case of high distributed power access, the accuracy of this algorithm will not be significantly affected.

(5) The fault location method based on this algorithm also occasionally makes location errors. The general positioning failure is related to the special situation of the line model, such as the line length being too short and the node having no load. However, the wrong location is still close to the actual fault line.

In general, through the verification of distribution network cases of different scales, it is proven that the method proposed in this paper eliminates the influence of a high proportion of DGs, and can accurately determine the location and type of faults. Compared with other algorithms, it is also verified that this algorithm has higher discrimination accuracy.

**Author Contributions:** Conceptualization, G.R. and B.J.; methodology, B.J.; software, X.H.; validation, B.J.; formal analysis, G.R.; investigation, G.R.; resources, X.Z.; data curation, X.Z.; writing—original draft preparation, G.R.; writing—review and editing, J.X.; visualization, K.T.; supervision, X.Z.; project administration, X.Z.; funding acquisition, G.R. All authors have read and agreed to the published version of the manuscript.

**Funding:** This research is funded by the Science and Technology Project of State Grid Jiangsu Electric Power Company, grant number J2021215.

**Institutional Review Board Statement:** Not applicable.

**Informed Consent Statement:** Not applicable.

**Data Availability Statement:** Not applicable.

**Acknowledgments:** The authors would like to thank the editors and anonymous reviewers for improving this paper.

**Conflicts of Interest:** The authors declare no conflict of interest.

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
