# Peer review of "Location of Multiple Types of Faults in Active Distribution Networks Considering Synchronization of Power Supply Area Data"

_applsci, doi:10.3390/app121910024_

Round 1
Reviewer 1 Report
This paper presents a method to detect line faults on distribution systems through measurements from intelligent devices such as micro PMUs. The paper is ridden with grammatical errors, so a thorough proofreading is suggested. For example, line 18, 31, 34, 41 on page 1. There are errors in the variables. For example, line 150, Yab_i ... at NODE i. The other concern I have is the use of DC power flow assumption, which is not valid for distribution systems. It is valid for transmission systems. The algorithm for fault detection is that of a meanshift method. It basically looks for a change in the measurements, which is prone to false alarms. It is not clear how the threshold Udz is selected.
Reviewer 2 Report
The authors of this article present analyzes related to the location of faults in lines and distribution networks. Proposed method for fault section location in active distribution network with area has been verified in the IEEE 37 network. The article emphasizes the need to perform such analyzes from the point of view of improving the operation of distribution networks.
However, there are some considerations and questions that should be addressed and clarified:
• In my opinion, the literature review is too cursory on the topic presented. Authors should review existing articles in more detail. Many works have been created on a similar subject.
• What is the voltage threshold value to determine if a short circuit has occurred?
• Did the authors perform analyzes for networks with a larger number of nodes?
• Line number 105 is: "T * fs" and should be: "T · fs" - in my opinion, the multiplication symbol is a full stop "·" and not the star "*".
• Figure 5, Figure 6, Figure 7, Figure 8 - What was the transition resistance during the simulation?
• In my opinion, the Authors should improve the presentation of the results obtained. In the "Results" section, there are no graphs with voltage and current values ​​for individual or selected calculation cases. The share of distributed generation sources should also be presented, since the authors wrote about it earlier.
• Fig. 5, Fig. 6, Fig. 7, Fig. 8 - no description (marking) of the horizontal axis. Font too small in charts. The text in the charts is hard to read.
• Figure 9, Figure 10 - too small font in the diagrams. The text in the charts is hard to read.
• The single letters "a" or the articles "the" appear at the end of some lines. They should appear at the beginning of the next line.
• Applications should describe exactly what is new in the presented article in comparison with other works of this type.
Reviewer 3 Report
The submitted manuscript aims at improving fault locating methods in ADN. However, following concerns should be addressed.
In this version of manuscript, some parts are improved. However,
1. Methodology is not presented in clear manner.
2. Literature review should be updated and the essence of fault locating in ADN should be highlighted. In this regard, some related papers are recommended.
10.1109/TSG.2022.3165388; https://doi.org/10.3390/s22072723
3. Results should be discussed more.
4. The authors are requested to revise the conclusion to meet the minimum requirements of conclusion part in a manuscript.
5. Moderate English changes required
Round 2
Reviewer 1 Report
After revision, the paper is still ridden with grammatical errors. It is unfortunate that in the very first reply in the author's response, the "before" and "after" edits are exactly the same. I will not attempt to list every single error here, but needless to say, there is still plenty. In addition, the idea of line outage detection from topology change has been studied extensively. The proposed idea of the paper is not novel. Consider the paper "Line Outage Detection Using Phasor Angle Measurements" by Tate. Their method also uses a mean shift. Here, the authors use the measured current instead of phase angle. Taking another measurement to detect outage as opposed to another variable that is a linear relationship to the former is not novel enough to warrant publication.
Reviewer 2 Report
Thank you for your responses and good luck.
Reviewer 3 Report
No further comments.